



# Regularities of new particle formation and evolution of existing atmospheric aerosol particles in a large (3200 m$^3$) isolated volume

Nikolay Romanov[1], Alexey Paley[2], Yuri Andreev[1], Sergey Dubtsov[3], Oleg Ozols[1], Yuri Pisanko[4], Dzhalil Sachibgareev[1], and Marina Vasilyeva[5]

[1]RPA "Typhoon", Obninsk
[2]State Oceanographic Institute, Moscow
[3]Voevodsky Institute of Chemical Kinetics and Combustion SB RAS, Novosibirsk
[4]Fedorov Institute of Aplied Geophysics
[5]Russian University of Transport (MIIT), Moscow

**Correspondence:** A. A. Paley (a_paley@mail.ru)

**Abstract.**

The paper reports on an investigation of nanometre-sized new particles formation (NPF) in aerosol-free outdoor air. This phenomenon was observed after filling of Large Aerosol Chamber (LAC) RPA "Typhoon" with the volume of 3200 m$^3$ with outdoor air, passed through HEPA 13 class filter (H13). During the summer-autumn period of 2018, even in the full darkness and in presence ionizing radiation only in the shape of secondary galactic cosmic rays, new particle formation with the particle size greater than 15 nm starts 0.5–1 hour after the end of LAC filling. During the 2018–2019 winter periods the NPF event was not observed once only.

Approximately one day after NPF narrow "bell-shaped" spectra with number concentration up to $10^4$ cm$^{-3}$ and mass concentration up to 0.6 µg per m$^3$ are formed. During the next five or more days, these size distributions evolve due to coagulation, while their asymptotic shape remains constant with relative count standard deviation $\sigma_c \approx 0.28$, and relative asymmetry $ras \approx 2$ ($ras = skewness/\sigma_c$). The value $ras \approx 2$ defines the analytical description of the size distribution as the gamma-distribution.

During additional purification of newly formed particles with the inner H13 filter, aerosol particles concentration in LAC decreases down to a few particles per cm$^3$. This concentration remained constant for more than a week. This demonstrates that new aerosol particles are formed by homogeneous gas-to-particle conversion of gaseous precursors, which passed through the external H13 filter. The mass concentration of newly formed particles depends on the concentration of precursors.

It was found that after filling LAC with outdoor unfiltered air, approximately after 10 hours the left-hand side of aerosol particle size distribution below 15 nm disappears, and after several days there forms an asymptotic "bell-shaped" size spectrum with $\sigma_c \approx 0.4$–0.5 and $ras = 2$–3. The modal diameter becomes about 150 nm after five days, while the size distribution greater than 200 nm remains unchanged. This allows concluding that aerosol particles greater than 200 nm have a life-time of more than five days, while particles smaller than 15 nm, not more than five hours.



The observed regularities of NPF and pre-existing aerosol spectra evolution may contribute significantly to understanding the processes of the formation of atmospheric aerosols, which are responsible for cloud and precipitation formation. They also should be considered during the design of purification methods for facilities and living spaces.

During the investigation of size distribution evolution of aerosol particles generated by the spraying of tap water, it was found that this aerosol particles size distribution transforms from a power law to a "bell-shaped" distribution in five days with $\sigma_c \approx 0.4$ and $ras \approx 2$. These results may be used for the development of aerosol evolution models.

*Copyright statement.* ©Author(s) 2020. This work is distributed under the Creative Commons Attribution 4.0 License.

## 1    Introduction

The atmospheric aerosol is quite necessary for the existence of a modern climate state (Lushnikov et al., 2015). The formation of cloud systems, which are the most important part of the recuperation of the evaporated water back to land, is impossible without atmospheric aerosols. Therefore, the understanding of atmospheric aerosol formation, evolution, and dissipation is one of the most urgent problems in the development of the planet's climatic models.

Today, after basic works of (Whitby, 1978) and (Rozenberg, 1983), it is considered (Baron and Willeke, 2001), that hydro-
scopically active atmospheric particles are formed by nucleating conversion gases (CG) into new particles (NP). These particles are further converted into accumulation mode with a count median diameter of 100–300 nm due to coagulation. This mode is the source of condensation nuclei for cloud particle formation. The review of (Kulmala et al., 2004) presents the situation with new particle formation (NPF) events for the period before 2000: "the development during the 1990s of new instruments to measure nanoparticle size distributions and several gases that participate in nucleation have enabled these new discoveries".
It follows from this review that NPF events are observed practically in all atmospheric conditions. Moreover, depending on meteorological and geographical conditions, their growth rate may vary by several orders of magnitude. The complexity and diversity of this problem are demonstrated in (Asmi et al., 2011), where aerosol spectra collected on 24 European stations are presented. This implies an understanding of the importance of a detailed study of the mechanism and conditions for the NPF.

Currently, this problem is being solved in two interconnected (parallel) directions. The first and main direction is to conduct
studies of the dependence of NPF processes in natural conditions, which differ in the ratio of anthropogenic and natural sources of CG. Note that to clarify the role of the latter in Finland in 1962 a station was established to study the ecosystem-atmospheric relationships for boreal coniferous forest (station for measuring ecosystem-atmosphere relation (SMEAR II)) (Hari and Kulmala, 2005). That forest covers $8\%$ of the Earth's surface and stores about $10\%$ of the total carbon in the terrestrial ecosystem. At this station for the period between 1996 and 2003 of observations an average explosive occurrence of NP was
recorded in $24\%$ of cases (Dal Maso et al., 2005). Similar processes were observed at other Northern field stations (Dal Maso et al., 2007) from which it was concluded that boreal coniferous forest, which releases organic matter in the form of isoprene and terpenes, participates in the formation of the aerosol component in the European part. In a subsequent paper (Kulmala





and Kerminen, 2008) an attempt is made to answer the question "How and under which conditions does the formation of new atmospheric aerosol particles take place?" And there they conclude that the predominant role of solar radiation in the

NPF mechanism from CG, as well as the possible role of ions. However, a detailed study of the NPF mechanisms in natural conditions is not possible due to the variability of these conditions. Therefore, the second natural method of cognition of such processes is their reproduction in laboratory (chamber) conditions.

Chamber conditions make it possible to investigate with controlled and repeatedly reproducible process parameters using a wide variety of accurate laboratory instruments to measure these parameters. However, this raises the question of taking into

account the influence of the walls of the chamber, in which, for small volumes, the processes of leaving formed particles on their walls can be a significant obstacle. Attempts to make low-cost chambers up to $240 \mathrm{~m}^3$ in volume from steel frames covered with a fluoroplastic film encountered the effects of the strong interaction of the material with organic gases (Matsunaga and Ziemann, 2010). But even using a similar chamber with a volume of $6 \mathrm{~m}^3$, specially made for operation under real conditions, important results were obtained on the influence of ozone and OH radical on NPF from terpenes secretions of pine trees,

specially grown for these purposes (Hao et al., 2009). Unfortunately due to the small volume of the chamber, the duration of the measurements was limited to $2 \mathrm{~hours}$. It should be noted that another important result obtained with the using of this chamber is the revealing of the amorphous nature of organic aerosol particles (Virtanen et al., 2010).

Using of a CLOUD (Cosmics Leaving Outdoor Droplets) camera built in CERN (Switzerland) with Proton Synchrotron with a volume of $26.1 \mathrm{~m}^3$ with walls of electro-polished stainless steel, a description of which along with the first results is in

the work (Kirkby et al., 2011), is found to be more suitable for obtaining quantitative results. In this chamber, a temperature from $183 \mathrm{~K}$ to $300 \mathrm{~K}$ can be controlled with an accuracy of $0.01 \mathrm{~K}$, and $373 \mathrm{~K}$ for cleaning the walls from contaminants can be created. The air in the chamber is created by the evaporation of liquid nitrogen and oxygen. A calibrated amount of conversion gases $SO_2$, $NH_3$, and other organic and inorganic gas components are added to the chamber. The ionic compound in the chamber is regulated from zero concentrations by creating a constant electric field to a natural background from galactic

cosmic rays and additional ionization of the air from the Proton Synchrotron $\pi^+$-meson beam. The solar radiation is simulated as well. Note that due to the processes of the influence of the walls, the duration of the studied processes in this chamber was limited to five hours. This required the constant addition of the studied gas mixture. From the results presented in this and subsequent works, the conclusion is drawn about the NPF mechanism as "ternary nucleation with ions playing a major but subdominant role" (Kirkby et al., 2016), (Dunne et al., 2016). It is also fixed that the compounds of CG are a complex system

of organic and inorganic molecules without a description of their specific form. Note that this chamber does not provide the injection of outdoor air.

Meanwhile, when studying the processes of NPF in various outdoor conditions, it shows the diversity of the CG composition. Thus it was shown (Pöschl et al., 2010) that in the Amazon region with no anthropogenic pollution, the major source of the condensing material is organic compounds, emitted by plants. Nevertheless, the ratio of biogenic and anthropogenic sources of

these precursors in the atmosphere is still poorly understood (Lehtipalo et al., 2018). This paper also states that this ratio will be very important in the future due to the decrease in emissions from fuel fossil combustion. The complexity of the composition of CG and the interaction mechanisms of individual components is indicated in a recent work (McFiggans et al., 2019), which





states "that formation mechanisms of secondary organic aerosol in the atmosphere need to be considered more realistically, accounting for mechanistic interaction between the products of oxidizing precursor molecules".

In connection with the above circumstances, it is very important to study the formation and evolution of NPF processes with CG in outdoor air in much larger volumes, where the influence of walls can be minimized. As such a volume, the question arises about the using of the 3200 m$^3$ Large Aerosol Chamber (LAC) available at the RPA "Typhoon", which provides for its filling with external air. Till recently, aerosol research was not possible due to the results of the research of (Smirnov et al., 2000). This work stated the existence of the "chamber" aerosol with the concentration of about thousands of particles per cm$^3$

in LAC and similar chambers. At present, the chamber is equipped with two (external and internal) HEPA13 aerosol filters (HEPA, 2019). As a result of these modifications for cleaning outdoor and indoor air from aerosols, it is possible now to reach almost zero aerosol concentration in LAC for a very long period (more than two weeks). It means that chamber walls and equipment, installed inside the chamber, do not produce aerosol particles, as was suggested earlier. The large volume of the chamber allows eliminating wall influence on particle sedimentation. Thus, in LAC it is possible to study processes of

atmospheric aerosol formation and evolution in close to natural conditions with obtaining asymptotic patterns. In particular, in LAC, the processes of nucleation formation of new aerosol particles from the introduced gas impurities can be experimentally studied with a trace of their evolutionary development, as well as the evolution of atmospheric aerosol in the absence of new particles source. The subject of this research is the experimental investigation of the mentioned above processes.

The obtained results can clarify some stages of aerosol dynamics in the real atmosphere. They may be also useful in the de-

sign of nano-aerosol filtration techniques in industrial and medical facilities. Nanoparticles may influence the human organisms as it is indicated, in particular, in (Youssefi and Waring, 2014).

## 2   Design of Large Aerosol Chamber (LAC) and description of experimental conditions

### 2.1   LAC device description

The largest cloud investigation chamber was built in 1954 in Texas, USA (Gunn and Paul, 1954). It has a spherical shape with an

internal diameter of 60 feet (volume 3200 m$^3$) with a steel wall thickness of one inch. It was first used to study the dependence of aerosol concentration on atmospheric aero-ion concentration (Phillips et al., 1955). Currently, publications on its use for meteorological purposes are not known to us. In Russia, a similar cylindrical chamber with the same volume 3200 m$^3$ was built in the town Obninsk in 1964. It is an integral part of the complex of facilities of RPA "Typhoon", designed for cloud and meteorological processes investigation. The first detailed description of its design and thermodynamic characteristics is given

in (Romanov and Zhukov, 2000). This description is cited, in particular, by authors on the use of poorly soluble hygroscopic substances for modification of warm clouds and fogs (Romanov et al., 2006), (Drofa et al., 2010) and regularities of cloud microstructure formation and evolution (Romanov and Erankov, 2013). However, due to the inaccessibility of the above work (Romanov and Zhukov, 2000) for English-speaking readers the description below partially duplicates the content of this work. It describes the possibilities of studying the properties of atmospheric aerosol, which has become possible recently due to the

retrofitting of its modern equipment.





At the beginning of this description, we note that in the study of cloud processes this chamber is referred to as the Big Cloud Chamber (Drofa et al., 2010). In aerosol research, it is often referred to as Large Aerosol Chamber (LAC). It is this name or just the chamber that is used in this work. The chamber represents hermetic steel cylinder $15$ meters in diameter and $18$ meters high, the inner surfaces of which are painted with ship paint. Outside, the chamber is covered with a shell of heat-insulating material $10$ cm thick and a decorative fence hiding its unaesthetic forms. The entire construction of the chamber is located in the building intended for the aforementioned complex and is surrounded by adjacent working premises isolated from each other. The wall thickness of the chamber is $6$ mm, which allows it to withstand excess pressures in $0.07$ MPa produced in about $50$ minutes by water-packed ring compressor. By reducing this pressure through a system of seven valves, a simulation of the formation a cumulus cloud with an adjustable rate of air mass rise is organised. Filling the chamber with outside air is carried out by sucking it through the windows of the premise adjacent to the chamber and the inlet using an exhaust fan located in the upper part of the chamber through a valve with a nominal diameter $DN = 800$ mm. In this case, the complete replacement of the internal air by the external environment takes place over a time of $\approx 0.5$ hours. Since 2014, it was possible to clean aerosols of air drawn in through the chamber by installing a HEPA13 (High Efficiency Particulate Air) class air filter at the entrance to the vestibule with dimensions equal to the dimensions of the entrance vestibule. At the same time, the time for replacing air in the chamber with the air cleared of aerosol increases up to about one hour. A similar air filter with its fan is also available inside the chamber. During its operation for $\approx 2$ hours, the aerosol concentration inside the LAC decreases by about three orders of magnitude.

Unfortunately, the appearance of the chamber cannot be represented using photographs, because, as mentioned above, it is located inside another building. Therefore, we have to confine ourselves showing its internal appearance, a photograph of which is presented in Fig. 1.

Note the presence in the chamber of an aero filter of class (High Efficiency Particulate Air) HEPA13 (position $4$) with dimensions equal to the dimensions of the entrance vestibule. As mentioned above, the second same air filter without a fan (external H13 air filter) can be installed on the outside of the chamber. The lanterns located on the walls of the chamber are used for commissioning and are usually turned off during experiments.

A description of the cloud droplet size meter developed by RPA "Typhoon" and located on suspension platform $3$ is available in (Drofa et al., 2010). To measure the size of condensation nuclei, they also used a laser particle size meter developed at RPA with a lower diameter limit $d_l = 0.2$ μm. In 2014, this device was replaced by SMPS Scanning Mobility Particle Sizer model 3936L88-N(TSI Inc.), based on the measurement of electro-mobility particles, which can be flooded with water vapour at a supersaturation of the order $0.5$ %. The size region used in these experiments was from $15$ to $1000$ nm with $115$ measurement channels. The air from LAC is sampled through a $2$ m long zinc coated steel tube with an inner diameter of $18$ mm. A similar tube for outdoor air sampling goes through a window near SMPS. The influence of these tubes on the measured particle size was considered during the data processing according to the SMPS manual.

**Figure 1.** Photo of LAC interior view. 1 – entrance portal with size $60 \times 160$ cm$^2$; 2 – rail for mounting the tested equipment; 3 – hanging platform with temperature sensors (for measuring dry and wet air), optical depth sensor with $4$ m optical path and a photoelectric sensor for measurement of cloud droplets size; 4 – inner HEPA13 filter.

## 2.2 Features of the interaction of the air with the walls of the chamber

The integral heat capacity of the walls and structures adjacent to them exceeds the integrated heat capacity of air by $10$–$15$ times. When simulate the cumulus cloud formation by reducing the previously created overpressure, the temperature difference between the wall and air usually varies from plus 5 to minus 10 K. In (Romanov and Zhukov, 2000), analytical relationships for the heat transfer rate in these processes are determined with the assumption of the convective mechanism of this process. Subsequent more detailed qualitative assessments of these processes show that the heat transfer consists of $3/4$ with IR emission





exchange of chamber walls with water vapours and $1/4$ convective heat exchanges. The convective heat exchange takes place
in form of ascending (descending) air flows in a 5–10 cm thick layer near vertical walls, and in form of horizontal movement
near the floor (Romanov, 2009). As a result, a homogeneous temperature is established in the entire horizontal layer of the
chamber (except for the regions near walls).

In stationary conditions without excess pressure, ideally, the air temperature in the chamber should coincide with the tem-
perature of its walls in the absence of any air movements. However, measurements using a GLL acoustic anemometer show
the presence of turbulent air flows at any time the chamber is in a closed state without any disturbing influences. An example
of such flows measured is presented in Fig. 2.

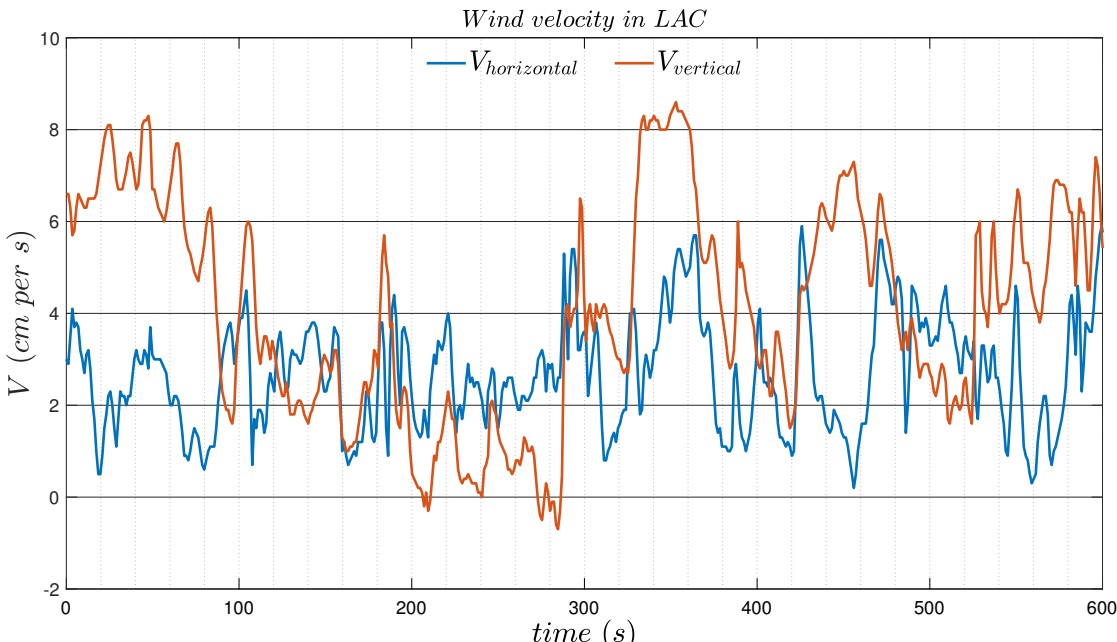

**Figure 2.** Vertical and horizontal wind velocity in LAC under stationary conditions

Note the chaotic air horizontal and vertical movements with the amplitude of several cm per s. Obviously, such movements
are induced by inhomogeneity of walls temperature due to their large dimensions. As a result of these movements, the aerosol
characteristics in horizontal level are averaged, so the measurement of aerosol spectra in one point represents averaged situation
for all chamber. This statement is confirmed by the fact that when the air is mixed in the chamber using a fan not indicated in
Fig. 1, the aerosol characteristics of the air practically do not change.





### 2.3 LAC radiation and ionization state

To study the new particle formation processes, it is important to know the concentration of aero ions, which can play a notice-
able role in their nucleation. However, according to the authors of the work (Wagner et al., 2017), the speed of new particle
formation (NPF) under their influence cannot exceed the speed of ion pairs (i.p.) formation under the influence of ionizing radi-
ation. For the chamber used in this work in CERN, this speed under the influence of natural radiation was $2$ i.p. per $cm^3$ per s.
To determine this characteristic, during the second half of 2019 and January 2020, several measurements series of $7$–$10$ days
of dose rate (DR) in LAC with a gamma radiometer using gas Geiger-Muller counters were carried out. It turned out that all
the results fit into the range $DR = 8$–$10\ \mu R$ per h. Hence, using the definition of X-ray $R = 2.06 \cdot 10^9$ i.p. per $cm^3$ per s, we
obtain the ionization rate in the chamber air at normal pressure $\approx 5$ i.p. per $cm^3$ per s. $DR$ measurements in the room near
LAC and outside the building usually exceeded the indication in the chamber by $\approx 20\%$. The explanation for a small decrease
in $DR$ in the chamber is that in the lower atmosphere the ionization of nitrogen and oxygen molecules is mainly responsible
for muons, which are formed when primary galactic protons collide with nitrogen and oxygen nuclei at altitudes of $10$–$20$ km
(Dorman, 2004). Highly penetrating muons are not filtered by LAC walls, but, at the same time, they can ionize gas molecules.
In this work, a detailed study of the aeroion formation is not conducted in the chamber, as it requires long-term experiments
with a variety of different parameters. A preliminary conclusion follows from several conducted experiments that the resulting
ion concentration in the chamber mainly depends on the aerosol concentration and is established within a few hours.

So for a single experiment with the aerosol concentration changing from $7 \cdot 10^3$ cm$^{-3}$ to $5 \cdot 10^1$ cm$^{-3}$ with using internal
H13, it was found that the concentration of light ions with mobility $\geq 0.4$ cm$^2$ per V per s was established for positive ions at
a level from $4 \cdot 10^2$ cm$^{-3}$ to $4 \cdot 10^3$ cm$^{-3}$ correspondingly.

### 2.4 Method of LAC cleaning from aerosol particles

At the beginning of this section, we note that the process of developing technology to achieve aerosol concentrations in LAC of
up to several particles per cm$^3$ has gone through several stages of understanding the process of formation and transformation
of aerosol. On the first stage after about one hour of LAC filling through the external HEPA filter, the air in the chamber is
almost completely (up to several tens in cm$^3$) substituted by filtered air mass from aerosol particles, greater than $15$ nm (this is
the lower limit SMPS $d_l$ for the used). However, it was found that after a while, new particles with the size near SMPS lower
detection limit appear. These particles then form a narrow "bell-shaped" distribution with concentration about $10^4$ cm$^{-3}$ and
with an obvious absence of particles from the small sizes side.

To clarify the origin of these particles, we used the procedure for removing this aerosol with an internal H13 aerosol filter.
At the same time, it turned out, after its operation for about two hours, that the aerosol concentration decrease almost to
zero (not more than a few particles per cm$^3$), which persists for an indefinitely long time. A more detailed description of
aerosol formation processes and size spectra transformation will be given below. Here, in Fig. 3 we give the pictures one of
the cycles. Dots refer to the beginning of $10$ minutes cycles of size spectra measurement start. The time before $t_1$ refers to the
measurements of outdoor aerosols.



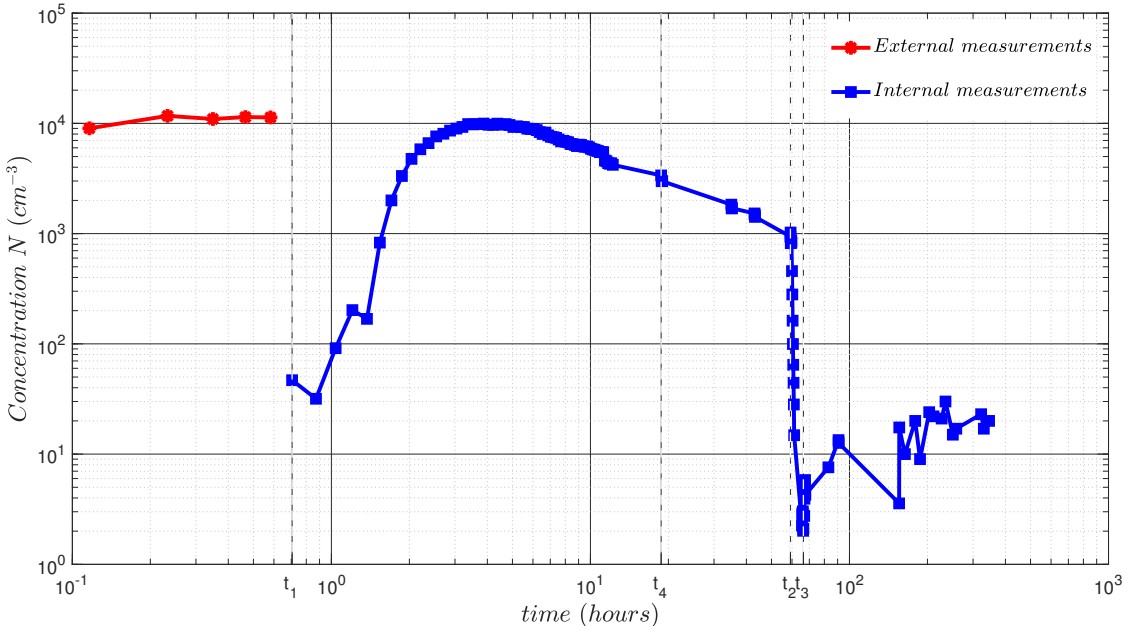

**Figure 3.** Time dependence of particle number concentration $N$ in LAC during its evolution after filling with outdoor air passed through external, and then through internal aerosol filters. AC – Cyc 18/02/19. Red circles – concentration aerosol particles after filling LAC with outdoor air. Blue circles- aerosol concentration inside LAC. $t1$ - time of the measurement start inside LAC after air cleaning with internal $H13$ filter. $t2$, $t3$ – time of the start and end of LAC cleaning with internal $H13$ filter. $t4$ – time moment of full size spectra registration.

It can be seen from Fig. 3 that particle number concentration decreases by more than two orders of magnitude after the external aerosol filter. Further increasing after about one hour can be explained by NPs, formed by nucleation of a specific gas component, passing through the external H13 aerosol filter. Naturally, the NP occurs at sizes below $d_l$. In the process of increasing their size due to Brownian coagulation, the particles become greater than the SMPS lower detection limit and are included in the total number of particles recorded by the device. This can explain a new particle concentration increasing.

Registration of the full spectrum with our equipment is possible only after $\approx 20$ hours of evolution (time moment $t_4$) when its left-hand side passes the border of $d_l$. At time moment $t_2$ switching on the internal aerosol filter for $\approx 2$ hours removes these new aerosols, and particle concentration in LAC decreases to $10\text{—}20$ particles per $\mathrm{cm}^3$. This concentration remains at this low value for more than $300$ hours (shown in Fig. 3).

        Such low concentration (almost zero), reached after the second filtration, was observed in many experiments. This low

concentration value remains almost constant for more than ten days. Switching on kitting, including agitating fan (not shown in Fig. 1), does not influence on aerosol concentration. It follows, that chamber walls and equipment inside do not generate aerosol particles.





## 2.5 The discussion of the results

The data presented above leads us to conclude that the material for NPF in the LAC is gas component contained in the outdoor
air, which will be called "conversion gas" (CG), passed through an external H13 filter. A specific property of this gas fraction
is its ability to completely transform into new aerosol particles without any external influences. We do not know the works
where such a phenomenon would be observed. In what follows we will call it "gas-particle conversion" (GPC). It was not the
knowledge of GPC phenomenon that led the authors of the work (Smirnov et al., 2000) to conclude that there is a chamber
aerosol in LAC. The erroneous statement obtained in this work about the existence of a chamber aerosol is based on the fact that
a corona discharge, used previously for LAC cleaning from aerosol particles, produces condensing material for new particle
formation itself, as shown in (Lapshin et al., 2012) and approved by our LAC experiments. We also note that the appearance of
new particles in enclosed spaces (Hirsikko et al., 2007), which the authors attribute to their generation by an unknown internal
source, is determined by GPC process.

In this paper, we will not discuss the mechanism of GPC due to the extreme complexity of this phenomenon. However, the
results obtained after research and evidence of its existence open up a huge number of research areas in solving meteorological
problems.

This paper presents the results of the following areas:

- Investigation of the possibility of an analytical description of the asymptotic spectra of NP in the process of their coagulation evolution. In this case, the dependence of the mass concentration of CG on the time of day and season is also
investigated (see Section 3).

- Investigation of the evolution of naturally occurring aerosol spectra in the absence of replenishment with new particles
(Section 4).

- Investigation of the asymptotic regularities of the spectra of other origins (Section 5).

Soon, the following works are planned:

- Investigation of aerosol particles spectra, produced in LAC by corona discharge in aerosol-free air.

- The study of the ability of various substances to NPF by the injection of test gas impurities into the aerosol-free LAC.

- Investigation of the dependence of the atmospheric ions concentration on the concentration of aerosols, which is regulated using an internal H13 filter.

## 3  Investigation of nucleation aerosol formation and evolution

As pointed above, after replacing the air in the chamber by passing through an external air filter, after some time, aerosol
particles begin to appear from the SMPS lower detection limit, which then evolves. Investigation of this phenomenon started
in 2015 after the installation of the H13 filter in LAC. Since that time 15 experiments in the summer-autumn period and 8 in





the winter period in 2018–2019 years were carried out. In almost all experiments (except 1 in the winter period) new particle formation has been observed. This confirms the idea that there is a constant source of aerosol precursor formation (Lushnikov
et al., 2015), (Baron and Willeke, 2001). The mechanism of their formation is not known yet (Yermakov et al., 2019).

Because of minimizing the attended factors, the investigation results of such a process can be as an experimental basis both for understanding the NPF mechanism and for developing theoretical models of this process. For this, the first subsection sets out the analytical definitions and relationships that will be used to present experimental results.

### 3.1    Data presentation format

According to coagulation and sedimentation laws rate of number concentration $N(t)$ change proportional to the second and the first degree of this quantity, where $K$ and $D$ are coagulation and sedimentation coefficient respectively (Fuchs and McDonald, 1964), (Anand et al., 2012), (Salimi et al., 2017), (Friedlander, 2000). This can be described by the differential Eq. (1).

$$\frac{dN}{dt} = -K \cdot N^2 - D \cdot N \tag{1}$$

The solution of Eq. (1) can be presented as:

$$N(t) = \frac{N(t_1) \exp\left((t_1 - t) \cdot D\right)}{\left[1 + K \cdot N(t_1) \Big/ D \cdot \left(1 - \exp\left((t_1 - t) \cdot D\right)\right)\right]} \tag{2}$$

Note, that in the work (Smirnov et al., 2000), the solution of the Eq. (1) is presented with a constant member on the right-hand side, which defines the particle inflow from an extraneous source. But we do not need such a solution, and therefore it is not given here.

When $D \to 0$, solution Eq. (2) can be transformed into a simpler form:

$$N(t) = \frac{N(t_1)}{\left[1 + N(t_1) \cdot (t - t_1) \cdot K\right]} \tag{3}$$

Note that expressions Eq. (2) and Eq. (3) imply that $K$ and $D$ do not depend on particle size. Typically they are used for $t \geq t_1$ to forecast dependence of $N(t)$. However, they may be used for time $t_0 < t < t_1$ of $N(t)$ behaviour reconstruction on the early stages of coagulation spectra formation, which cannot be measured by instruments. Value $t_0$ can be defined from the condition when denominator in expression Eq. (3) is zero.

$$t_0 = t_1 - \frac{1}{K \cdot N(t_1)} \tag{4}$$





Obtaining a more complex expression for $t_0$ based on expression Eq. (2) does not make sense because it will lead to the same numerical value using the limit transition procedures.

The above expressions will be used for comparison of experimentally measured $N(t)$ dependencies with theoretical ones. For a more detailed description of experimental size distributions we will use not only commonly used distribution function
$n(d)$ (Baron and Willeke, 2001), but normalized distribution function $f(d) = n(d)/N$ moments. This $m(n)$ moments are described by integrals Eq. (5), in which absent integration limits from $0$ to $\inf$ just for writing simplicity.

$$m(n) = \int f(d) \cdot d^n \, dd \tag{5}$$

At that $m(0) = 1$, and count median diameter $d_m = m(1)$. Other relative dimensionless parameters are defined by the following expressions:
Relative count standard deviation :

$$\sigma_c{}^2 = \frac{\int f(d) \cdot (d - d_m)^2 \, dd}{d_m^2} \tag{6}$$

Asymmetry coefficient ($kas$):

$$kas = \frac{\int f(d) \cdot (d - d_m)^3 \, dd}{\left[ \int f(d) \cdot (d - d_m)^2 \, dd \right]^{3/2}} \tag{7}$$

Relative asymmetry ($ras$) is another useful parameter:

$$ras = kas \Big/ \sigma_c \tag{8}$$

Equation (6) and Eq. (7) may be presented in the form of second and third combination moments respectively.

Let's consider the most simple and widely used two-parameter distribution functions for experimental data approximation.These are the gamma ($g$), the log-normal ($ln$) and the Smirnov ($Sm$) distributions. Parameters of these distributions are explicitly defined by $d_m$ and $\sigma_c$ values and have the following relation between $ras$ and $\sigma_c$ (Romanov and Erankov, 2013),
(Tammet and Kulmala, 2014):

$$ras(g) = 2; \quad ras(\ln) = 3 + \sigma_c{}^2; \quad ras(Sm) = 4 \Big/ \left(1 - \sigma_c{}^2\right) \tag{9}$$

It follows that the gamma distribution has the smallest relative asymmetry $ras = 2$, which is independent of other distribution parameters. The form of the gamma distribution is given below:

$$f_\gamma(u) \, du = \frac{\mu^\mu}{\Gamma(\mu)} \cdot u^{\mu-1} \cdot \exp(-\mu \cdot u) \, du \tag{10}$$

Here $u = d/d_m$; $\mu = 1/\sigma_c{}^2$. When $ras > 2$ one can use a sum of other distributions, with weights, defined by Eq. (9).



## 3.2 Experimental results on aerosol nucleation mode formation and evolution at positive temperatures

At a qualitative level, it was found that new particles are characterized by a large variety of mass concentration and narrow size distributions during the summer-autumn period. Special attention has been paid for this period due to its importance to the understanding of new particle formation processes. The period of the cyclonic synoptic situation on August 30th, 2018 was

studied in detail, when LAC was filled the air mass with a temperature of $20\,^\circ$C and $12\,\mathrm{g\,per\,m^3}$ of water content through an external HEPA filter at $12\,\mathrm{h}\,17\,\mathrm{min}$ local time. The investigation of new particle formation and evolution was carried out for six days. The obtained experimental results are presented in Fig. 4–7.

### 3.2.1    Formation and evolution of new particles size distributions

Figure 4 shows the time dynamic of aerosol spectra. The size distribution of outdoor aerosols with the concentration equal to

$N_{ext} \approx 4 \cdot 10^3\,\mathrm{cm^{-3}}$ is marked with $external$. Index $t = 0$ corresponds to the start of measurements, and this time moment coincides with the end of LAC filling with the purified air. The aerosol concentration at that moment was $N_0 = 27\,\mathrm{cm^{-3}}$. Since at this moment the particles were registered only in separate channels, the curve $t = 0$ is presented as separate marker points. Next indexes refer to the period in hours from the end of the chamber filling to the time moment when the measurement started.

Figure 4 shows that after $1/3$ hours, the formation of additional aerosol with size above the detection limit in $15\,\mathrm{nm}$ is observed on the background of residual aerosol. The almost complete size distribution is detected after three hours, and after $20$ hours there are no particles less than $15\,\mathrm{nm}$. This time moment $t = 20$ hours may be considered as the point for comparison of experimental data with Eq. (2) and Eq. (3). The measured distributions at $t = 3$ hours may use as qualitative data, as the only small part of left-hand side of size distribution is below the measurement limit.

The next figure shows the time dependence of integral parameters of size spectra in Fig. 4 with additional spectrum in $117$ hours.

From Fig. 5b it follows that despite a wide range of changes in average diameter (from $30$ to $73\,\mathrm{nm}$) during the period from $3$ to $147$ hours, the values of the relative count standard deviation $\sigma_c$ and asymmetry $ras$ do not change significantly. The value of $\sigma_c$ asymptotically tends to $0.27$–$0.29$. The value of $ras$ is positive and fluctuates around the value $2$. These values,

together with Eq. (9), indicate that the gamma distribution is most suitable for new particle size spectra approximation. The experimental and approximation spectra comparison are presented in Fig. 6.

One can see from Fig. 6 that the gamma distribution Eq. (10) is in a good agreement with the shape of experimental size distributions.

### 3.2.2    The evolution of particle number concentration

Results of the measuring of number and mass concentration related to Fig. 4 are presented in this section.

Lines with marks in Fig. 7a represent measured concentration in the range from $15$ to $1000\,\mathrm{nm}$. Blue solid line is $N_2(t)$ calculated from Eq. (2) with reference point $N(t = 20) = 4200\,\mathrm{cm^{-3}}$ and with coefficients $K = 9.05 \cdot 10^{-6}\,\mathrm{(cm^3\,per\,hour)} \equiv$





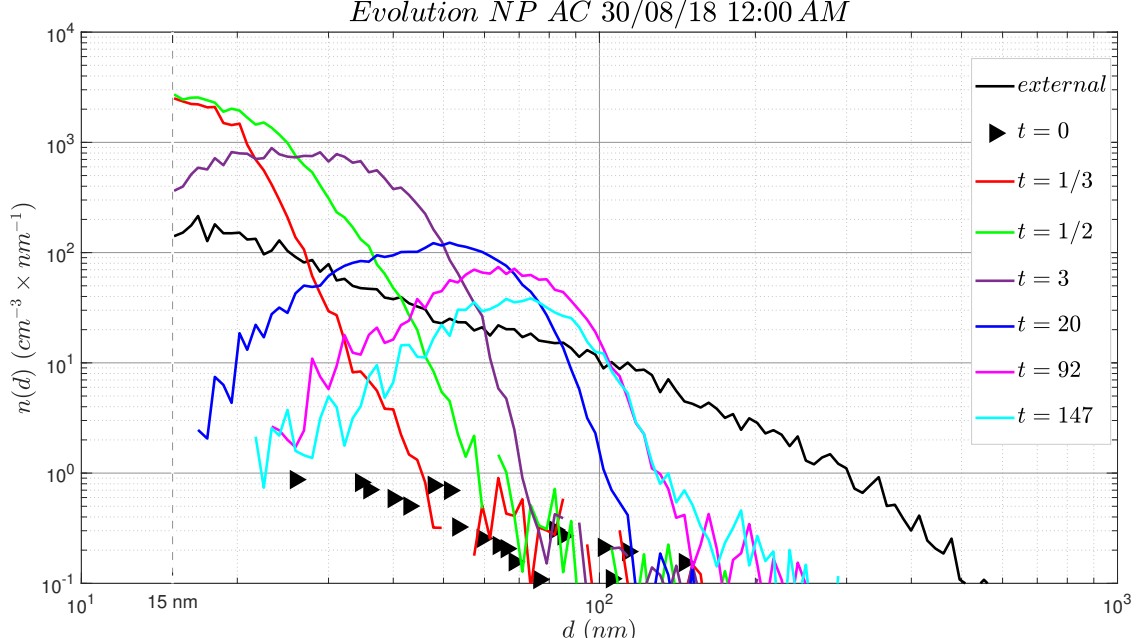

**Figure 4.** Formation and evolution of the secondary aerosol in LAC after filling it with purified air by external $H13$ filter on August 30th, 2018. Curves correspond to the start time of $10$ minutes measuring periods, counted from the moment the end of LAC filling. The vertical dashed line is the lower detection limit of SMPS. $Ext$ – size distribution of outdoor aerosol particles, $t = 0$ – aerosol particles size distribution, measured after LAC filling with purified air. $t = 1/3$, $t = 1/2$, etc. – time moment (in hours) of size distribution measurement start after LAC filling

$2.5 \cdot 10^{-15}$ $(\mathrm{m^3\,per\,s})$ and $D = 2.6 \cdot 10^{-3}$ $(\mathrm{hour^{-1}}) \equiv 7.2 \cdot 10^{-7}$ $(\mathrm{s^{-1}})$, which were estimated by successive iterations. The discrepancy between calculated and experimentally measured values (including $t = 3$ h) does not exceed $1.5\,\%$. The dependency

$N_3(t)$, which was calculated by Eq. (3) with the same coefficients $K$ and $N$ ($t = 20$ hours) for the dependency estimation of the last member in Eq. (1), is presented by the red line. The insignificant difference between $N_2(t)$ and $N_3(t)$ one can make from the qualitative comparison. Increasing $N_3(t)$ over $N_2(t)$ takes place in $t = 3$ hours and quantitatively has the value in $12\,\%$ as well as decreasing takes place in $t = 160$ hours and has the value $25\,\%$.

     The mass concentration $m$ was calculated from experimental data, assuming that particles are spherical, with a density

equal to $1\,\mathrm{g\,per\,cm^3}$. The dashed line in Fig. 7b represents these values. As seen from this figure, the mass concentration value increases by about $10\,\%$ for the time from 3 to 20 hours. This increasing can be explained in Fig. 4, where one can see, that no particles less $15$ nm are fixed. At the same time, for the evolution time $\geq 90$ hours, the value decreases by about $2.5$–$3$ times (the aerosol mass should be constant in the coagulation process). This so large decreasing cannot be explained by sedimentation, which is defined by the last member in the Eq. (1), as follows from the comparison is given above. For clarity,





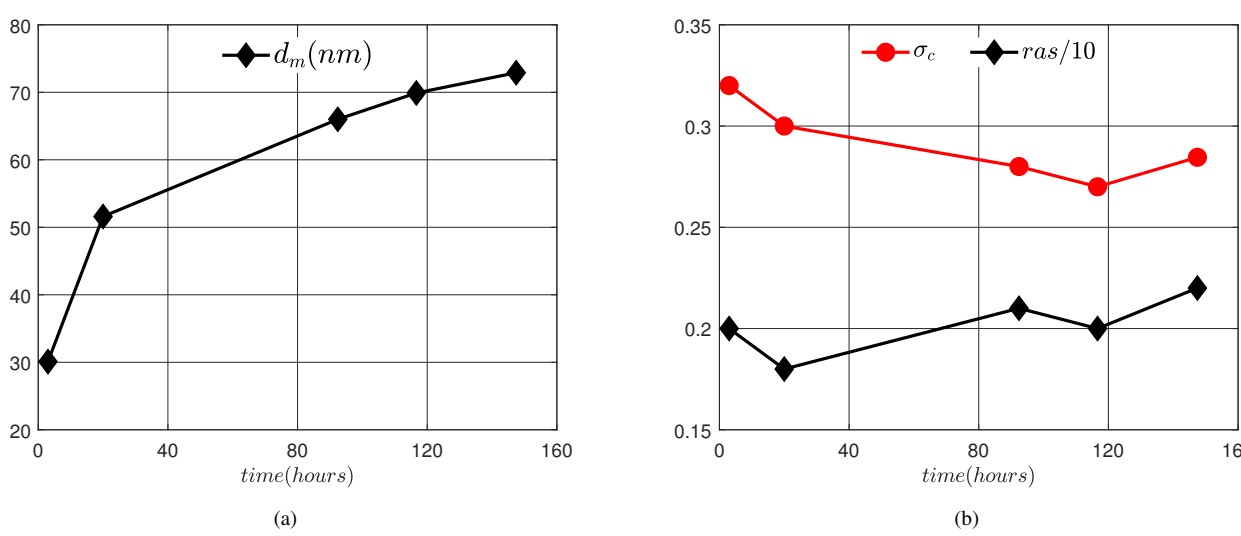

| (a) | (b) |

**Figure 5.** Time dependence of (a) count median diameter $d_m$; (b) relative count standard deviation $\sigma_c$ and relative asymmetry, divided by factor of 10 $ras/10$.

the dependency $m_2(t)$, which takes into account sedimentation, is shown in the same Fig. 7b, is calculated by Eq. (11) with the same coefficients $K$ and $D$.

$$m_2(t) = m(t_0) \cdot \frac{N_2(t)}{N_3(t)} \tag{11}$$

Here, $m(t_0 = 20h) = 0.44\ \mu\mathrm{g}\ \mathrm{per}\ \mathrm{m}^3$.

The comparison of the data in this figure shows that under the above assumptions about the sphericity and constancy of the

NP density taking into account the sedimentation although leads to some decrease in the measured and calculated difference in mass NP concentration, but does not explain the very large its value (about in $2.5$ times for the five days).

The considerations may be explained by the following hypotheses:

– aerosol ageing, resulting in its structure compaction;

– evaporation of aerosol products in the time scale of several days;

– overestimation of the small particle size with SMPS.

It is clear that the considerations expressed above are hypothetical in nature however in the current state of NP nature knowledge, one will have to dwell.



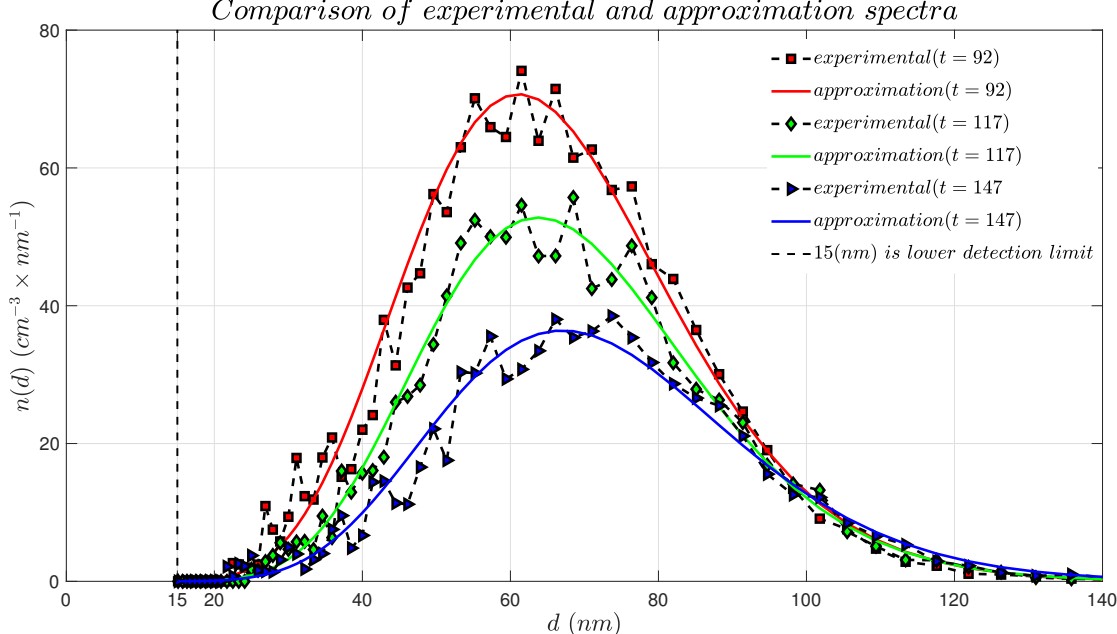

**Figure 6.** Comparison of experimentally measured (lines with marks) and calculated by Eq. 10 (lines) aerosol size distributions after 92, 117, and 147 hours of their evolution. The vertical dashed line is lower detection limit of SMPS.

### 3.2.3 Dependence of NP characteristics on day time

The measuring of the left-hand side of the size distribution (15 nm) has never vanished. It means that there is a source of
condensing products at least of hourly availability. One of these sources may be plant emissions. The efficiency of such a source
should depend on day time. To check this hypothesis, experiments on aerosol evolution in purified from aerosol particles air
was carried out in September 2018. The chamber was filled with air in the evening and night time. Results of such experimental
data are presented in Table 1.

The data, presented in Table 1, shows that maximal aerosol concentration ($0.6\,\mathrm{\mu g\,per\,m^3}$) of conversion gases, which is
determined by the NP mass, is observed in air, which was sampled into LAC at day time. When air is sampled in the evening
or at night, the NP concentration decreased to $0.2\,\mathrm{\mu g\,per\,m^3}$ and $0.09\,\mathrm{\mu g\,per\,m^3}$ respectively. Next, we can note the almost
complete coincidence of the $ras$ values in the first and last two rows of Table 1. The overestimation of this value for the
second row can be attributed to insufficient time delay. So, the values $\sigma_c = 0.28$ and $ras = 2$ may be considered as universal
asymptotic values and used in modelling of formation and evolution of conversion aerosol. The size distribution of the formed
particles is narrow, therefore the estimated above sedimentation ($5\,\%$) will not influence on aerosol size spectra, and the gamma-





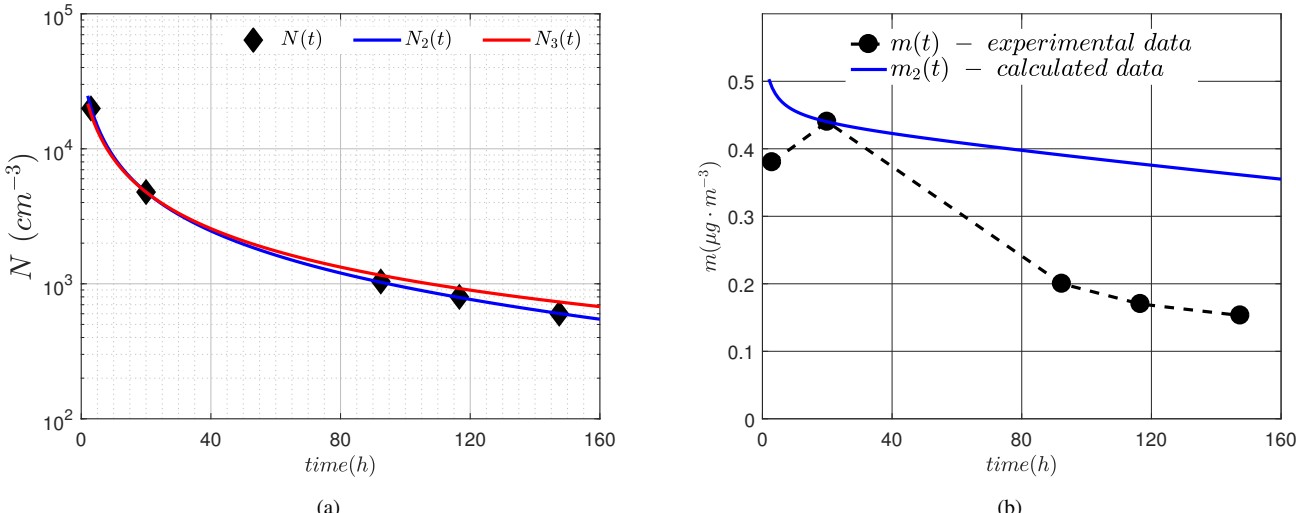

(a)             (b)

**Figure 7.** Comparison of experimentally measured number concentration $N(t)$ and calculated $N_2(t)$ (blue line), and $N_3(t)$ (red line) concentrations time dependencies. (b) - Experimental $m(t)$ and calculated $m_2(t)$ dependences of mass NP concentrations during their evolution in LAC.

**Table 1.** Time dependence of characteristics for outdoors aerosol and aerosol filtered air in LAC in the summer-autumn season. Date – date of air sampling. Time – the local time of the end of chamber filling. a – water content

| | | Outdoor air | | | Air in LAC | | | | |
|---|---|---|---|---|---|---|---|---|---|
| Date | Time | N $(cm^{-3})$ | m $(\mu g \cdot m^{-3})$ | a $(g \cdot m^{-3})$ | $\Delta t$ (h) | $d_m$ (nm) | m(NP) $(\mu g \cdot m^{-3})$ | $\sigma_c$ | ras |
| 30/08/18 | 12:00 am | $4.2 \cdot 10^3$ | 7.36 | 12 | 20 | 52 | 0.44 | 0.30 | 1.9 |
| 05/09/18 | 6:00 pm | $2.0 \cdot 10^4$ | 15.5 | 12.2 | 16 | 39 | 0.2 | 0.33 | 2.8 |
| 06/09/18 | 1:00 pm | $1.1 \cdot 10^4$ | 8.6 | 9 | 20 | 57 | 0.58 | 0.28 | 1.8 |
| 18/09/18 | 10:30 pm | $7.3 \cdot 10^3$ | 4 | 7.9 | 20 | 33 | 0.09 | 0.25 | 2.4 |

distribution, Eq. (10) may be assumed as the asymptotic function for the coagulation of NP. Fixed in Table 1 the lowest NP mass concentration at night may support the hypothesis, that NP is formed from the volatile plant emissions.



### 3.3 Experimental results of aerosol in winter time

Winter aerosol should differ from summer aerosol as noted above. The difference is primarily defined by the absence of a
biogenic CG source, which is plant emission. A similar effect was observed during the investigation of NP formation in the
Antarctic (Jokinen et al., 2018), in which new particles were detected when the air masses were coming from the open ocean,
and there were no particles detected when the air masses were coming from mainland or ice fields. The second difference is
low absolute air humidity value at negative temperatures. These low values remain after air warming in LAC, while relative
humidity decreases significantly.
We studied four situations during winter 2019 when there was solid snow in the region, gentle breeze, and stable negative
temperature. The results of these studies are presented in Table 2.

**Table 2.** Dependence of characteristics of outdoor aerosol and aerosol, produced in LAC from filtered air, on the time $\Delta t$ in winter period.
Date – date of air sampling. Time – sampling time.

| | | Outdoor air | | | | Air in LAC | | | | |
|---|---|---|---|---|---|---|---|---|---|---|
| Date | Time | T ($°C$) | N ($cm^{-3}$) | m ($\mu g \cdot m^{-3}$) | a ($g \cdot m^{-3}$) | $\Delta t$ (h) | $d_m$ (nm) | m(NP) ($\mu g \cdot m^{-3}$) | $\sigma_c$ | ras |
| 10/01/19 | 00:00 pm | -11 | $1.4 \cdot 10^3$ | 1.2 | 2.2 | 22 | - | 0.00 | - | - |
| 11/02/19 | 00:30 pm | -1 | $7.2 \cdot 10^3$ | 10 | 4.2 | 20 | 31 | 0.08 | 0.33 | 2.5 |
| 14/02/19 | 00:00 pm | -7 | $10 \cdot 10^3$ | 6 | 2.5 | 31 | 34.5 | 0.08 | 0.30 | 3.0 |
| 18/02/19 | 10:00 pm | -5 | $2.1 \cdot 10^4$ | 3.8 | 1.9 | 35 | 42 | 0.095 | 0.34 | 2.9 |

The first episode on 10/01/19 with low negative temperature corresponds to the edge of an anticyclone with isothermal
temperature profile in 300 meters layer after small snowfall. The number and mass concentration of outdoor aerosol was low.
During this situation, there was no NP formation in LAC for 22 hours of observation. The other episodes refer to relatively
low negative temperatures with sufficiently large number and mass concentrations of aerosol. They differ in initial temperature,
water content and time on LAC filling, while characteristics of the formed in LAC particles with mass concentration less than
$0.1 \, \mu g \, per \, m^3$ are almost similar.

### 3.4 Results discussion

Before the discussion, it should be noted that the process of LAC filling with air filtered by H13 filter, passed CG, takes about
one hour. The process of new particle formation from CG, their coagulation and coagulation with pre-existing in LAC aerosol
particles, as well as their and CG removal from LAC takes place during the filling. Without going into complex computational


models of these processes, it is possible to accept the moment 20 minutes before the filling end as a start time of the aerosol evolution in LAC. However, to avoid uncertainties arising from this, we accept the moment of the filling end as a start time.

Another point is that at the moment of measurement start with SMPS, new particle formation from CG is most likely
finished. This conclusion follows from the fact, that just after the appearance of the size spectra mode, the measured articles concentration starts to decrease in time. It means, that in fact, we observe coagulation of already formed new particles.

From the results presented in Fig. 5b and Fig. 6, it can be seen that, in less than a day after the start of the NP formation process, their spectra assume an asymptotic form that coincides with the gamma distribution. Note that this conclusion we have for the first time. Indeed, the formation and asymptotic behaviour of coagulation spectra is the subject of numerous theoretical
studies (Fuchs and McDonald, 1964), (Friedlander, 2000). One of the last publications on the subject is (Anand et al., 2012). However for atmospheric aerosols, the theoretical study of such dependencies is complicated by uncertainty in the description of the coagulation nucleus for specific situations. Until now, the modelling of such processes has been complicated by the influence of small chamber walls. An illustration of this approval is the work (Seipenbusch et al., 2008), in which for a chamber of 2 m³ the results of experimental and theoretical studies of the platinum aerosol evolution with an average diameter of about
7 nm in a clean chamber and the presence of a simulating atmospheric aerosol are given. Obtained in this work the results of aerosol coagulation up to sizes of about 100 nm for a time of up to 250 minutes had to be obtained with continuous operation of the aerosol source due to diffusion of the aerosol onto the walls. An important direction in studying the properties of the atmospheric aerosol is the fitting of analytical ratio for approximating the size distribution function, which is the critical aerosol characteristic in modelling both cloudy and general climatic processes. The question of the analytical expression of aerosol
spectra was studied in detail in (Tammet and Kulmala, 2014), where two-power law four-parameter distribution is proposed. The results are obtained with the participation of the authors, who provided the measurements of aerosol spectra. It also points out many previously proposed models, one of which is the sum of three lognormal distributions (John, 2011).

The above results show that, at least for NP formation, the gamma distribution is most suitable.

## 4  Evolution of the outdoor aerosol in LAC

The asymptotic behaviour of aerosol size spectra in the absence of its source is a scientific interest. The understanding of the atmospheric aerosol evolution in living space or working area, closed for outdoor air, is the practical interest. Therefore we conducted several experiments to study the evolution of various types of atmospheric aerosols in LAC. The outdoor aerosol is sampled into LAC with exhaust fan till the moment when the size spectra of outdoor aerosol and aerosol in LAC coincide. LAC is closed for more than 7–10 days after filling. Special port in LAC with HEPA filter is used to compensate for internal
and external pressure.

### 4.1  The aerosol spectra time evolution for various isolated air masses

Spectra evolution for four different air masses is presented in Fig.8a – Fig. 8d.





**Figure 8.** The result of aerosol spectra evolution in air masses isolated in LAC. Curves correspond to the period in hours from the moment of isolation. AC, Cyc and Dish – an anticyclone, a cyclone, and a dish.

The presented data show that at the moment of measurements start, there are particles smaller than 15 nm. After about 10 hours they coagulate with the formation of particles greater than the lower detection limit. After about 24 hours aerosol size distribution transforms into a "bell-shaped" distribution with a maximum at about 70–100 nm. During the next five days the average particle diameter grows to 100–200 nm, while the width of the size distribution narrows, at the constant right-hand side. After this time, up to the maximum observed 15 days, the decreasing of the spectrum is observed.





Time dependences of relative parameters $\sigma_c$ and $ras$ of the above presented spectra are illustrated in Fig. 9a – Fig. 9d.

**Figure 9.** Time evolution of the relative parameters for aerosol size distributions, presented in Fig. 8a – Fig. 8d

From the data in Fig. 9, it follows that relative asymmetry value is constant during all the time of observation. It is close to
$ras = 3$ for an anticyclone (Fig. 9a), and about $ras = 2$ for a cyclone, a dish, and an anticyclone with smoke. The value of $\sigma_c$ reaches its asymptotic value of $0.4$–$0.5$ after approximately $100$ hours of aerosol evolution for all presented cases.



## 4.2 Time dependence of number and mass concentration of outdoor isolated aerosol

One can see, that for the presented in Fig. 8 and Fig. 9 data, isolated in LAC aerosol size distribution parameters change significantly during their evolution. From theoretical studies (Friedlander, 2000) follows that the coagulation coefficient (core) depends greatly on the particle size ratio. This makes almost impossible to solve coagulation equations analytically. Numerical solving of the coagulation problem also possesses great uncertainty, because coagulation core depends greatly on particle shape. And finally, even the solutions found will not always be worth the effort spent on them due to the diversity of natural situations. As was demonstrated in the previous section, the assumption on the constant value for the coagulation coefficient can describe $N(t)$ for narrow enough size distributions. Therefore we studied the possibility of using simple Eq. (2) and Eq. (3) to describe the evolution of outdoor aerosol. It turn out that for the presented in Fig. 8 $N(t)$ dependence may be described by Eq. (3) with two constant values for coagulation coefficient $K < 24$ and $K > 24$, for the period less than one day and more than one day respectively. These values presented in forth and fifth columns of Table 3. From the data in Table 3 one can see that coagulation coefficients values are almost the same for experiments in 2014 (a, b and d), while for the year 2018 experiment (c), they are about two times higher. This may be due to the difference in atmospheric aerosols for these years. We are not present a graphical comparison of experimentally observed and calculated from Eq. (3) with coefficients from Table 3 values because practically the parameters fitting should be performed for each meteorological situation separately. Note, that the difference between observed and approximated values does not exceed $5\,\%$ in the time scale from one to about ten days.

Equation (3) implies that particle mass is constant with time. However, the experimentally measured mass concentration $m$ decreases significantly (about $1.5\,\mathrm{times}$ for five days). Equation (2) also cannot describe the time behaviour of aerosol number and mass concentration, even at $t > 24$ hours. The obtained data do not allow suggesting time dependence of $K(t)$ and $D(t)$, therefore we offer the following expression to describe experimentally observed $m(t)$:

$$m(t) = \frac{m(t_1)}{\left[1 + \alpha \cdot (t - t_1)\right]^{6/5}} \tag{12}$$

A similar dependence for the number concentration was given in (Voloshchuk, 1984) with some suggestions about coagulation nuclear. This expression has no physical ground, but it is simple. The values for $\alpha$ are shown in the last column of Table 3. The discrepancy of experimental and calculated from Eq. (12) values in Table 3 does not exceed $10\,\%$ for the whole time range at $t_1 = 24$ hours.

Taking into account the individual nature of the coefficient $\alpha$ values are given in Table 3, it can be noted that the coefficient $\alpha$ is useful for the aerosol mass evolutionary changes estimation in a specific large volume.

## 5 Experimental results on WC-50 aerosol evolution

One of the nanoparticle generation methods is spraying of salt aqueous solutions solution, upon drying of which nanosized particles are formed. LAC has ideal conditions to study the evolution of such aerosol. On the other hand, the interpretation





**Table 3.** The values of the number concentration at the initial moment $N_0$ and after 24 hours $N_{24}$; $K$ - coagulation coefficient in the period before and after 24 hours; mass concentration $m_{24}$ and coefficient $\alpha$ in Eq.(12) for data, presented in 8a–8d.

| Number | $N_0$ $(\mathrm{cm^{-3}})$ | $N_{24}$ $(\mathrm{cm^{-3}})$ | $K < 24$ $(\mathrm{m^3 \cdot s^{-1}})$ | $K > 24$ $(\mathrm{m^3 \cdot s^{-1}})$ | $m_{24}$ $(\mathrm{\mu g \cdot m^{-3}})$ | $\alpha$ $(\mathrm{h^{-1}})$ |
|---|---|---|---|---|---|---|
| a | $10 \cdot 10^3$ | $4.3 \cdot 10^3$ | $2.0 \cdot 10^{-15}$ | $1.4 \cdot 10^{-15}$ | 10 | $4.9 \cdot 10^{-3}$ |
| b | $11 \cdot 10^3$ | $3.7 \cdot 10^3$ | $2.1 \cdot 10^{-15}$ | $1.5 \cdot 10^{-15}$ | 7.7 | $5.5 \cdot 10^{-3}$ |
| c | $12 \cdot 10^3$ | $3 \cdot 10^3$ | $3.0 \cdot 10^{-15}$ | $2.5 \cdot 10^{-15}$ | 10 | $9 \cdot 10^{-3}$ |
| d | $25 \cdot 10^3$ | $6.5 \cdot 10^3$ | $1.4 \cdot 10^{-15}$ | $1.0 \cdot 10^{-15}$ | 25 | $4.5 \cdot 10^{-3}$ |

of evolution regularities for the aerosol particles with known composition and compact structure is more reliable than for the new particles with unknown composition and structure. The aerosol particles produced by water-packed ring compressor WC-50, used for the production of the excess pressure in LAC, were used in our first experiments. When the authorized filter is installed in WC-50, there are no aerosol particles produced by the compressor. To produce aerosol particles in our experiments, the filters were bypassed. The concentration of the major components in the sprayed water was the following: Ca (1100), Na (400), Mg (375), Sr (200), and Fe (13) (mg per l).

The evolution of aerosol after its sampling into LAC is presented in Fig. 10.

Figure 10 shows, that $n(d)$ has a mode at about 30 nm at the moment of the measurements start, and a power-law dependence for $d > 100$ nm. The position of spectra maximum shifts to the right with time, and the shape of size distribution becomes a "bell-shaped" with the maximum at about 150–200 nm. Table 4 presents the aerosol characteristics in detail.

It is seen from Table 4, that relative count standard deviation values stabilize after the first day and reaches a value of $\sigma_c \approx 0.4$, and relative asymmetry tends to the value about 2 after three days. So, the gamma distribution may be used for the sprayed aerosol particles. Note, that the decreasing of mass concentration, which was observed in the above described experiments, is also was observed for these aerosol particles.

## 6 Conclusion

The major results of this research may be divided into two parts. The first one is that the upgrading of Large Aerosol Chamber (LAC) of RPA "Typhoon" with the volume of $3200 \, \mathrm{m^3}$ with HEPA 13 filters, made it possible to study gas-to-particle formation processes. New particle formation was observed after filling LAC with filtered air by an external filter. A natural process leading (explaining) to the appearance of new particles in the LAC under conditions of complete darkness and background concentration of aeroions formed by secondary galactic cosmic radiation is the homogeneous nucleation of conversion gases (CG), which are not retained by the H13 filter. Next removing of these particles with the internal H13 filter results in the





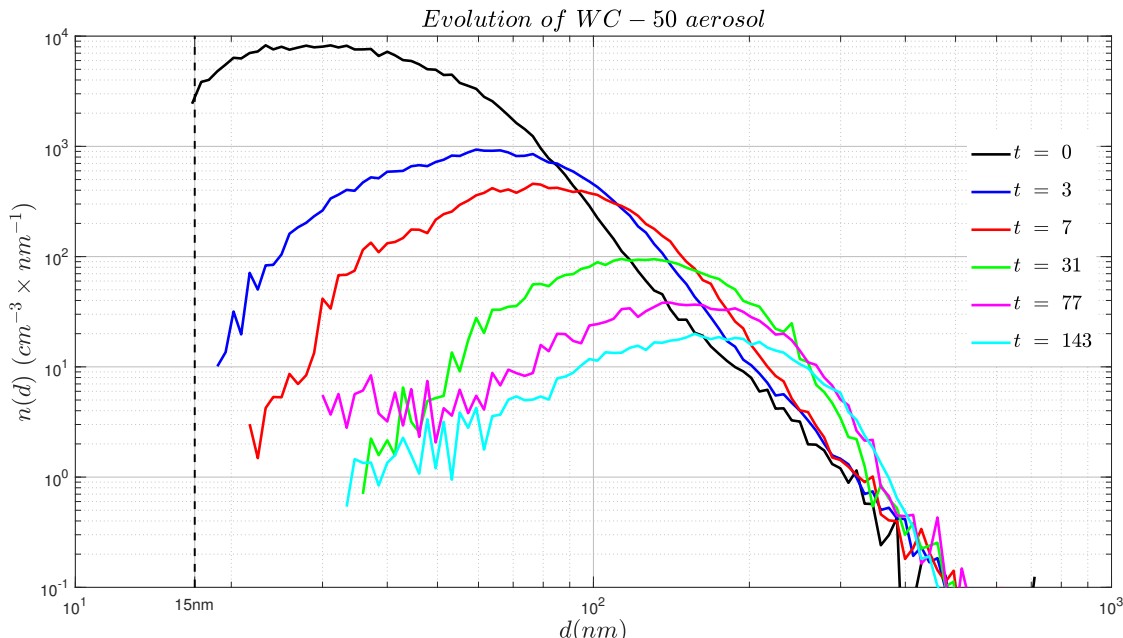

**Figure 10.** Size distributions $n(d)$ of aerosol from the compressor. $t = 0$, $t = 3$, etc. refer to evolution time in hours. The vertical dashed line is the lower detection limit of SMPS.

**Table 4.** Time dependence of the parameters of sprayed aerosol spectra.

| t (h) | N $(\mathrm{cm}^{-3})$ | m $(\mu g \cdot m^{-3})$ | $d_m$ (nm) | $\sigma_c$ | ras |
|---|---|---|---|---|---|
| 0 | $3.23 \cdot 10^5$ | 30 | 43.4 | - | - |
| 0.96 | $1.18 \cdot 10^5$ | 28 | 60.5 | - | - |
| 2.96 | $6.49 \cdot 10^4$ | 28 | 77 | 0.45 | 6.0 |
| 6.58 | $3.76 \cdot 10^4$ | 30 | 95.4 | 0.43 | 5.4 |
| 31.4 | $1.18 \cdot 10^4$ | 29 | 146 | 0.38 | 3.04 |
| 77 | $5.91 \cdot 10^3$ | 24 | 171 | 0.40 | 2.12 |
| 143 | $3.38 \cdot 10^3$ | 17.5 | 188 | 0.39 | 2.06 |





practically zero aerosol concentration in LAC for a very long (more than a week) period. On the one hand, this fact indicates that all the amount of CG, pumped in LAC, is the source of NP. On the other hand, this fact indicates that the chamber walls

and installed in the chamber equipment do not generate aerosol particles. So, after secondary removal, it makes possible to study processes of aerosol formation and evolution in well controlled conditions. Large chamber volume allows for avoiding significant particle wall and floor sedimentation. The data presented above indicate the necessity to use the two-stage procedure for cleaning of working areas, with a time gap enable for gaseous precursors to form new particles, which are removable by H13 filter.

The second part of the opportunities pointed above is the realisation the aerosol formation and evolution processes investigation. The process of homogenous aerosol formation and evolution due to gas-to-particle conversion has been studied. It was found that almost full spectrum with sizes greater than SMPS lower detection limit ($d = 15$ nm) appears on the coagulation stage in 2–3 hours after filling LAC the purified outdoor air. At the same time, the NP concentration exceeds the value of about $10^4$ particles per cm$^3$ and very quickly lowers due to Brownian coagulation. Naturally, in the period of new particles nucle-

ation that is inaccessible for complete measurements, their concentration far exceeds the value indicated above, which, in turn, almost an order of magnitude exceeds the background concentration of aeroions, equal to $(2-4) \cdot 10^3$ per cm$^3$. Therefore, we can say that in our case aeroions do not play the main role in the NPF process. A similar statement exists in the work (Kirkby et al., 2016), where it is said that the aeroions do not play the main role in the NPF process.

The NP spectrum of this process of homogenous aerosol formation and evolution due to gas-to-particle with the constant

mass concentration transforms in the narrow ''bell-shaped'' spectrum with the concentration $\approx 3 \cdot 10^3$ per cm$^3$ about a day later. During the next days (five or more) these size spectra transform due to Brownian coagulation, while their shape remains constant. The relative count standard deviation $\sigma_c$ for the summer-autumn period is about $0.28$, and relative asymmetry $ras \approx 2$. The last value determines the analytical approximation of this asymptotic form as the shape of a gamma distribution.

The NP mass concentration calculated by measured spectra stays almost constant during the first two days, which allows

correlating it with the mass CG concentration. The mass concentration calculated by measured aerosol spectra (in the assumption that specific density equals $1$ g per cm$^3$) varies from $0.09$ to $0.6$ μg per m$^3$ for night and day time respectively during the summer-autumn period. For winter conditions, it does not exceed $0.1$ μg per m$^3$ and does not depend on the time of a day. The decrease in aerosol concentration at night is probably caused by the decrease in plant emissions. Note, that new particle formation at night was observed in Australia during episodes when air masses came from the ocean (Salimi et al., 2017).

The second research was to investigate the aerosol spectra evolution of outdoor (not filtered) air, isolated in LAC. It was found that particles smaller than $15$ nm disappear after $7$–$10$ hours in typical meteorological situations. Asymptotic a "bell-shaped" curve with $\sigma_c \approx 0.4 - 0.5$ and $ras = 2 - 3$ formed in LAC after several days of evolution. Spectra with value $ras \neq 2$ can be approximated by the sum of the gamma and the log-normal distributions with appropriate value weight coefficients for measured $ras$. After about five days the modal diameter reaches the value of about $150$ nm, while the part of

the size distribution for particles greater than $200$ nm has the same shape. It follows, that particles smaller than $15$ nm are less than $10$ hours old, and in anticyclone conditions with the wind velocity about $1$ m per s, these particles are formed at the distance, not more than $40$ km. So, they are formed from local sources of gaseous precursors. Such particles smaller than





15 nm have been observed not only in our research (Fig. 8) but in 24 European stations (Asmi et al., 2011) also. In all these cases, a noticeable concentration of aerosol of about 15 nm was recorded. It means that such gaseous precursors are present in

the whole continental part of Europe. Particles greater than 200 nm are older than five days, it means that they are formed in remote regions.

The third research was to investigate the aerosol evolution, generated by spraying tap water with known chemical composition. It was found that the shape of size distribution such aerosol transforms from a power law to a "bell-shaped" with $\sigma_c \approx 0.4$ and $ras \approx 2$ for five days. Thus, the gamma distribution is also an asymptotic shape of the aerosol spectra. These results may

be used for the simulation of various aerosol evolution models.

Finally, we do not discuss the composition of gaseous precursors in our research. It should be noted that the results of our research indicate that there is a constant process of gas-to-particle formation in the summer-autumn period in Europe (even in the winter).

# 7 Acknowledgments

The authors thank the employee of the RPA "Typhoon" V. Yakhryushen for measuring radiation characteristics, former employee RPA V. Erankov for ensuring the operation of the LAC equipment and C. Philipchenko for help with preparing graphics in MATLAB and text in LATEX.

The reported study was partially funded by RFBR and Novosibirsk region according to the research project $\#19 - 43 - 540009$.



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
