# Peer review of "Regularities of new particle formation and evolution of existing atmospheric aerosol particles in a large $(3200 \text{ m}^3)$ isolated volume"

_Atmospheric Measurement Techniques, 2020_

## Referee Comment (RC1) · Anonymous Referee #1 · 23 Nov 2020

Review of AMT submission "amt-2020-172" by Romanov et al. titled "Regularities of new particle formation and evolution of existing atmospheric aerosol particles in a large (3200 m3) isolated volume"

Significance: The study introduces a large environmental chamber facility "Large Aerosol Chamber (LAC) RPA ("Typhoon")" located in Obninsk, Russia. It has been previously used for cloud physics investigations, and now, is considered to be used for studies of atmospheric new particle formation. In this publication several observations of particle dynamics inside the LAC are described, presumably due to the want to include this very large chamber setup into the international collection of atmospheric

simulation chambers serving the atmospheric community. I certainly recognize the need for various different platforms for studying atmospheric transformations, and generally, the larger the facility, the rarer it is. This chamber is unique already for being so massive, and thus could perhaps supply key insight into atmospheric particulate matter formation, permitting to study the nucleation processes with minimal influence from prevailing surfaces. However, there are several practical issues that should be resolved before I could advocate its utilization within the community. Moreover, the current level of presentation is not in line with the common norm of AMT articles, and thus I suggest major revisions to the structure and context, before I can propose this article to be accepted for publication in AMT.

Major scientific concerns: 15 nanometers can hardly be discussed as "new particle formation". A lot needed to happen that the particles could reach this size range. Also, it is clear that the particle formation, if there is such, is initiated already when filling the chamber, as atmospheric particle growth rates very seldom reach 15-30 nm/hour, which seems to be needed to explain the current observations. Or is the whole observation just secondary organic aerosol growth on top of already present small particles? With this instrumentation we will simply not know.

It seems strange to have instruments inside the chamber facility if you are attempting to measure such a fragile process as gas-phase nucleation. I do recognize that this might have been needed for the cloud formation experiments which I don't possess expertise, but to study new particle formation (=nucleation) in a chamber with instruments inside seems strange indeed. For me it is very difficult to see how these machines inside the chamber would not influence the particle formation experiments. I have to ask: In what other chambers studying nucleation of nanoparticles there are instrumentation inside the chamber? Again, nucleation is a very fragile process indeed.

To the same background influence, one has to ask what does the "ship paint" contain that has been used to paint the inner walls? Is this a possible source of nucleation precursor chemicals to the chamber? Commonly investigations go through a lot of trouble

in ensuring the walls behave inert. Yet, I definitely agree on what is said about Teflon not being an optimal choice. I do not know what this "ship paint" is composed of, but it seems like a rather questionable surface for conducting NPF studies - which are notoriously influenced by even very minor concentrations of certain chemicals. According to Figure 1, the chamber is literally full of potential particle sources, and I'm not sure if you could ever decide with the current instrumentation if there's new particle formation due to them or not. In connecting this to your presented results, you imply a 15 nm/hour growth rate in a very large chamber without background particles and without oxidants? (Well of course there are

oxidants such as OH and Criegee radicals, as the chemistry is initiated by ozonolysis reactions – yet none of this is discussed in the paper, which is another problem). This is a very large growth rate and needs a large fraction of condensing species present, and a steady source of small seed particles. I guess what I'm really trying to say here is: I don't see why this paper claims to report new particle formation, as the measurement had no techniques capable of extracting this information. Yet, the study of aerosol dynamics in forming the larger particles seems certainly worthy of reporting! However, that steps out of my key knowledge, and some better qualified person should judge on the novelty of the results. So, the last line of the current abstract "These results may be used for the development of aerosol evolution models" is to my mind, absolutely to the point.

To sum it up: this is an interesting and rather unique facility for studying atmospheric aerosol dynamics at sizes above 15 nm, with the current instrumentation and description. I don't see the ability to study new particle formation without significant further studies and serious updating of the major instrumentation.

Specific suggestion: I strongly recommend trying to measure the aerosol precursor pool of the chamber – what are the main gaseous molecules inside of the chamber after filling? Especially what are the SO2 and O3 concentrations in your experiments – as these gases are most likely needed to induce the first steps of the particle formation

in a dark chamber. For hydrocarbons and their oxidation products, a proton transfer reaction mass spectrometer (PTRMS) and gas chromatograph (GC) would suffice to get an idea about the constituents oxidizing and ultimately forming particulates. For the actual in-situ aerosol precursors you would need to have an atmospheric interface equipped chemical ionization MS applying some of the clustering reagent ions (e.g., $NO_3-$ and $I-$). Like it's said in the abstract "The mass concentration of newly formed particles depends on the concentration of precursors." Absolutely! And thus, it would be important to know about them to make statements about new particle formation vs secondary mass growing on top of small existing particulates.

Major formatting concerns: Currently the level of presentation is a bit awkward, and the manuscript in parts reads more of a historical documentary, or a thesis, than a scientific article. There are too much description of previous works and platforms, which are not really needed to describe the current facility or the obtained results. For a historical description a review article would be more suitable format, yet then also the journal would probably have to change. To my mind AMT is reserved for introducing and describing new methodologies and instrumentation, and as such, this facility would fit the description. Obviously, also some background is needed from the previous work to understand the context, yet this should be done briefly enough to not confuse the main topic of the work. The language also has to be improved substantially before this draft is really ready for submission. Ideally a native English speaker should edit the text first. I am not one and thus I cannot perform this task. Currently, I am not 100% sure I understand the text exactly as it was supposed to, and I presume my non-nativity was part of the reason why I could understand the text quite well already.

Few specific comments: I did not dwell on the minor problems within the text as I found the text has to be revised significantly before it can be considered for publication in AMT. However, I have picked here just a few issues which could be fixed already at this stage. Somewhat uncommon terminology is used throughout the text. Some examples: *"conversion gases" seems uncommon. Maybe the wording you are looking

for is "trace gases"? *What is "skewness"? *What is the meaning of the sentence: "The value ras ≈ 2 defines the analytical description of the size distribution as the gamma distribution."? *"The large volume of the chamber allows eliminating wall influence on particle sedimentation." To my mind "sedimentation" specifically relates to gravity, and is thus different from wall deposition? *It wasn't clear to me where this came up from: "This allows concluding that aerosol particles greater than 200 nm have a life-time of more than five days, while particles smaller than 15 nm, not more than five hours." While I understand the comment on 200 nm particles, I did not understand how did you come up with 5-hour lifetime for smaller than 15 nm particulates if they were never measured?
* * *

---

## Author Comment (AC1) · 4 Dec 2020

The authors express gratitude to the Reviewer for useful comments. Below are our responses to the Reviewer's comments:

[Figure]

**1 Comment #1**

**1.1 Reviewer comment**

$15$ *nanometers can hardly be discussed as "new particle formation". A lot needed to happen that the particles could reach this size range. Also, it is clear that the particle formation, if there is such, is initiated already when filling the chamber, as atmospheric particle growth rates very seldom reach $15 - 30$ nm/hour, which seems to be needed to explain the current observations. Or is the whole observation just secondary organic aerosol growth on top of already present small particles? With this instrumentation we will simply not know.*

**1.2 Response**

Our experiment was as follows. The Large Aerosol Chamber with air inside was plugged hermetically. You can see in Fig.3 particle number concentration of filled air was $10^4$ particles per cm$^3$. After that, the air inside the Chamber was filtered so that no particles greater than $10$ nm (technical characteristic of HEPA (HEPA, 2019)) remained inside the Chamber. After that, we began to measure the particle size spectrum inside the Chamber with the help of a particle size spectrometer mounted outside the Chamber. The particle number concentration of all detected particles was less than $50$ particles per cm$^3$. We observed the increase of particles with sizes more than $15$ nm 3 hour after the start of the measurements (particle number concentration increased to a maximum of $10^4$ particles per cm$^3$). After that, we filtered the air once more so that no particles greater than $10$ nm remained inside the Chamber and started measurements, but no increase of the particles' concentration was observed during $300$ hours of probing. We concluded that observed new particles of $15$ nm scale had originated from precursors due to the gas-to-particle conversion or had been condensed on surfaces of particles smaller than $10$ nm (both had been contained in the air) before the second filtering, and we did not observe the increase after the second filtering because almost all precursors and particles smaller than $10$ nm had been already involved into the process of gas-to-particle transformations and had been swept out as condensation centers of new particles by the second filtering. Such low concentration (almost zero), reached after the second filtration, was observed in many experiments. This low concentration value remains almost constant for more than ten days.

**2 Comment #2**

**2.1 Reviewer comment**

*It seems strange to have instruments inside the chamber facility if you are attempting to measure such a fragile process as gas-phase nucleation. I do recognize that this might have been needed for the cloud formation experiments which I don't possess expertise, but to study new particle formation (=nucleation) in a chamber with instruments inside seems strange indeed. For me it is very difficult to see how these machines inside the chamber would not influence the particle formation experiments. I have to ask: In what other chambers studying nucleation of nanoparticles there are instrumentation inside the chamber? Again, nucleation is a very fragile process indeed. To the same background influence, one has to ask what does the "ship paint" contain that has been used to paint the inner walls? Is this a possible source of nucleation precursor chemicals to the chamber? Commonly investigations go through a lot of trouble in ensuring the walls behave inert. Yet, I definitely agree on what is said about Teflon not being an optimal choice. I do not know what this "ship paint" is composed of, but it seems like a rather questionable surface for conducting NPF studies -which are notoriously influenced by even very minor concentrations of certain chemicals. According to Figure 1, the chamber is literally full of potential particle sources, and I'm not sure if you could ever decide*

*with the current instrumentation if there's new particle formation due to them or not.*

**2.2 Response**

The instruments inside the Chamber were turned out and were not enabled in our experiment. After the Chamber had been purified with the help of an internal HEPA filter (in Fig.3 the time interval $t_2 - t_3$), the concentration of aerosol particles decreased by $3$ orders of magnitude and remained low (about $30 - 40$ particles per cm$^3$) during the next $300$ hours. This testifies against the chamber walls (including its "ship paint") and inactive instrumentation mounted inside the chamber as potential sources of precursors or condensation centers.

The measurements provided in the one point of LAC, far from any equipment and wall. As shown in Fig.2, air velocity enough small to influence the experiments. The huge volume of LAC allows to research particle evolution in almost natural conditions.

**3 Comment #3**

**3.1 Reviewer comment**

*I don't see why this paper claims to report new particle formation, as the measurement had no techniques capable of extracting this information. Yet, the study of aerosol dynamics in forming the larger particles seems certainly worthy of reporting! However, that steps out of my key knowledge, and some better qualified person should judge on the novelty of the results. So, the last line of the current abstract "These results may be used for the development of aerosol evolution models" is to my mind, absolutely to the point.*

[Figure]

**3.2 Response**

In our paper, we describe the experimental study of the evolution of aerosol particles in the presence of a source of condensable products and propose an analytical description of asymptotic coagulation spectra of aerosol particles. To us, the obtained experimental data provide evidence that gas-to-particle conversion processes take place in the Large Aerosol Chamber. Certainly, we cannot specify a mechanism of new particle formation in the Chamber without the gas composition (ozone, nitrogen oxides, hydrocarbons, etc.) analysis, which becomes the subject of our forthcoming publications.

The article presents the evolution of ultrafine particles with a size of more than $15$ nm, including NPF, under almost natural conditions, to the size of cloud condensation nuclei, which is very important for assessing the effect of submicron particles on meteorological processes in the atmosphere.

**4 Comment #4**

**4.1 Reviewer comment**

*Major formatting concerns: Currently the level of presentation is a bit awkward, and the manuscript in parts reads more of a historical documentary, or a thesis, than a scientific article. There are too much description of previous works and platforms, which are not really needed to describe the current facility or the obtained results. For a historical description a review article would be more suitable format, yet then also the journal would probably have to change. To my mind AMT is reserved for introducing and describing new methodologies and instrumentation, and as such, this facility would fit the description. Obviously, also some background is needed from the previous work to understand the context, yet this should be done briefly enough to not confuse the*

*main topic of the work. The language also has to be improved substantially before this draft is really ready for submission. Ideally a native English speaker should edit the text first.*

**4.2  Response**

The referee drew attention to the increased volume of chamber descriptions. This section has been expanded on the advice of an APC referee. The main argument is that the materials of Russian studies are poorly represented in the English-language literature and are not known to a wide range of specialists. In particular, the search for literary sources that we referred to for the description of the equipment used caused difficulties for the reviewers. Undoubtedly, AMT has extensive experience in analyzing the informative attractiveness of article materials. And, if the editorial board finds the shortest material, we will revise the article, considering all the recommendations.

**5  Comment #5**

**5.1  Reviewer comment**

*"conversion gases" seems uncommon. Maybe the wording you are looking for is "trace gases"?*

**5.2  Response**

The term "conversion gases" denotes the gas components that participate in gas-to-particle conversion; not all "trace gases" do participate in the process.

**6  Comment #6**

**6.1  Reviewer comment**

*What is "skewness"?*

**6.2  Response**

Skewness refers to asymmetry of statistical distribution. If the curve is shifted to the left or the right, it is skewed. Skewness can be quantified as a representation of the extent to which a given distribution varies from a normal (statistically symmetric) distribution. A normal distribution has zero skewness while a lognormal distribution, for example, exhibits some degree of right-skew. See, for example, (Teegavarapu, 2019).

**7  Comment #7**

**7.1  Reviewer comment**

*The large volume of the chamber allows eliminating wall influence on particle sedimentation." To my mind "sedimentation" specifically relates to gravity, and is thus different from wall deposition*

**7.2  Response**

Of course, it is a typo. The sentence should be: Large chamber volume allows for avoiding significant particle wall deposition and floor sedimentation.

**8 Comment #8**

**8.1 Reviewer comment**

*This allows concluding that aerosol particles greater than* $200$ nm *have a life-time of more than five days, while particles smaller than* $15$ nm*, not more than five hours. "While I understand the comment on* $200$ nm *particles, I did not understand how did you come up with* $5$-hour *lifetime for smaller than* $15$ nm *particulates if they were never measured?*

**8.2 Response**

Life-time for $200$ nm particles was evaluated from experimental data to be more than $5$ days, this estimate is close to that reported by various authors, for example (Kreidenweis, 1999). The Figure 8 (Fig. 8a Fig. 8d) shows that during the time interval between the two first measurements of particle size spectra (which is $5$ hours in average) the concentration of $15$ nm particles decreases by an order of magnitude.

**References**

HEPA, https://vozdyx.ru [Online; accessed 01-December-2019], 2019

Teegavarapu, Ramesh S. V., *Trends and Changes in Hydroclimatic Variables : Links to Climate Variability and Change*, https://doi.org/10.1016/B978-0-12-810985-4.00001-3, pp. 1-98, 2019

Kreidenweis, S., G. Tyndall, M. Barth, F. Dentener, J. Lelieveld and M. Mozurkewich, *Aerosols and clouds in Atmospheric Chemistry and Global Change*, pp. 117-154, 1999

---

## Referee Comment (RC2) · Anonymous Referee #2 · 22 Dec 2020

This manuscript describes a Large Aerosol Chamber (LAC), with the volume of 3200 m3. The chamber is located at the RPF "Typhoon" facility in Obninsk, Russia. The LAC is a massive environmental chamber, which has previously been used for cloud and meteorological process studies and is evolving to be used to study aerosol processes including new particle formation. The results of several experiments of particle production and growth are also presented.

General comments

Such a massive chamber is indeed unique, and there is certainly is a need for facilities that allow the study of atmospheric chemical processes with minimal wall influences. I

have some specific concerns or questions regarding its use for these types of experiments, which are outlined below. My main comment is that in its present state, the organization and use of English in the manuscript is not appropriate for publication in AMT. The authors even state that one of the reasons for writing this manuscript is that a description and use of the facility has been previously published, but there is a lack of accessibility for English readers. The poor English in the text made it so that in some parts the point could not be understood. A significant rewrite and review by someone knowledgeable with English grammar is necessary. In looking at the growth results, I don't really see anything new understanding from this work. The results rely SMPS measurements and are basically just confirmation of aerosol physics.

In the description of the LAC, it says that the walls we coated with "ship paint". Further description of this "ship paint" is necessary. As an atmospheric chemist, I find the use of a painted surface for the wall of an environmental chamber concerning. Have any tests been performed to confirm that the walls are not a source of hydrocarbons. If the authors hope that this facility will be used to study new particle formation (NPF), a chemical process, the off-gassing of the walls needs to be shown not to be a source of contamination or interference.

Along the lines of chemistry, I am surprised at the lack of chemical measurements. The authors simply use the term constituent gas without going into detail as to its makeup. A few simple instruments such as an ozone, SO2, and NOx monitors would add greatly to the understanding of the constituent gas and the subsequent chemistry it undergoes. A gas chromatograph (GC) would give a good idea of the reactant hydrocarbons are. These are all more or less standard off the shelf measurements yet would add greatly to understanding the chemical makeup of the precursor species. I would also make a sincere effort at changing the sampling strategy and move the instruments to the outside of the chamber. I find it hard to believe that the presence of the instruments and equipment shown in Figure 1 inside the chamber don't affect the chemistry.

The authors state that new particle formation is observed and then present particle size

distributions with a 15 nm lower cutoff. New particle formation involves the formation of particles in the 1 nm size range. 15 nm particles are actually quite large compared to newly formed ones and have undergone a wide range of possible processes from condensation growth to coagulation. There are instruments capable of measuring size distributions down to 1 nm (particle size magnifier and like instruments) and such instruments would be necessary to talk knowledgably about NPF and initial growth processes.

Has any thought been given to the addition of a photolytic source? Most NPF is driven by photolytic processes.

My overall impression is that the LAC has the potential to be a useful addition to the environmental chambers available to the atmospheric community. For this to happen, more chemical measurements will need to be added. The field of aerosol science, especially that of particle formation has evolved to include chemistry and the transformation and heterogeneous interactions of gas phase precursors. One has only to look at the CLOUD chamber to see the potential impact the LAC could achieve. I feel that if the experiments discussed were repeated with the inclusion of the above suggested measurements and the results discussed in terms of chemistry (oxidation, ozonolysis, formation of low volatility species, etc) this work would be a strong addition to AMT (with a rewrite from a native English speaker). However, in its present state I cannot recommend that this manuscript for publication.

---

## Author Comment (AC2) · 26 Dec 2020

**Response to the Reviewer**

December 26, 2020

The authors express gratitude to the Reviewer for useful comments. Below are our responses to the Reviewer's comments:

**1    Comment #1**

**1.1    Reviewer comment**

*The results rely SMPS measurements and are basically just confirmation of aerosol physics.*

**1.2    Response**

We understand the importance of studying atmospheric chemical processes.

We would also like chemists not to underestimate the importance of studying the physical processes occurring in the atmosphere. Ultimately, aerosols formed because of

chemical processes will only then have a significant impact on meteorological processes in the atmosphere when, as a result of their evolution, they grow to the size of cloud condensation nuclei.

**2 Comment #2**

**2.1 Reviewer comment**

*My **main** comment is that in its present state, the organization and use of English in the manuscript is not appropriate for publication in AMT. The poor English in the text made it so that in some parts the point could not be understood.*

**2.2 Response**

Many of the article authors took examinations at Oxford. Also, the article text was prepared with the participation of specialists who are native speakers of the English language.

**3 Comment #3**

**3.1 Reviewer comment**

*In the description of the LAC, it says that the walls we coated with "ship paint". Further description of this "ship paint" is necessary. If the authors hope that this facility will be used to study new particle formation (NPF), a chemical process, the off-gassing of the walls needs to be shown not to be a source of contamination or interference.*

**3.2 Response**

The new particle formation is not the subject of this article.

We will undoubtedly use the advice of the referee when we investigate the new particle formation.

Studying the possible gas formation from the walls and from the equipment located in the chamber to make sure that their influence on the processes under study is negligible is methodologically unprofitable. Gaseous components do not directly affect the investigated processes of particle evolution. Their influence can only be felt when new particles are formed from them. It is guaranteed to cover by measurements all theoretically possible gas components that can form new particles. It is too difficult a task. We methodically solved this problem much easy. We experimented. And we checked whether new aerosols were formed in a hermetically sealed chamber due to all possible gas contamination from the walls and equipment inside the chamber. The experiment showed that during $300$ hours of observation, a negligible number of new particles (no more than $30$ particles per $cm^3$) are formed in the chamber. And this is more than two orders of magnitude less than the number of particles whose evolution we have studied. The article presents of this experiment results (see Fig. $3$, where $t > t_3$).

**4 Comment #4**

**4.1 Reviewer comment**

*Along the lines of chemistry, I am surprised at the lack of chemical measurements.*

**4.2 Response**

The article does not consider the chemical processes of the aerosol particle formations.

**5 Comment #5**

**5.1 Reviewer comment**

*I would also make a sincere effort at changing the sampling strategy and move the instruments to the outside of the chamber.*

**5.2 Response**

In our experiments, the spectrometer is installed outside the chamber.

**6 Comment #6**

**6.1 Reviewer comment**

*I find it hard to believe that the presence of the instruments and equipment shown in Figure 1 inside the chamber don't affect the chemistry.*
**6.2   Response**

The article did not study the issues of chemical processes. Therefore, we did not investigate whether the presence of the tools and equipment shown in Figure 1 inside the chamber affects chemistry or not.

The article presents the experiment results, which shows that the equipment installed inside the chamber does not contribute to the new particle formation. Accordingly, it does not affect the investigated processes of the particle evolution formed from the external atmosphere air.

**7   Comment #7**

**7.1   Reviewer comment**

*There are instruments capable of measuring size distributions down to 1 nm (particle size magnifier and like instruments) and such instruments would be necessary to talk knowledgably about NPF and initial growth processes.*

**7.2   Response**

The study of the chemical processes of the new particle formations and their initial growth is not included in the list of issues under study.

The article presents material on the study of the growth of already formed submicron particles. It is shown that these particles during their evolution, even in the absence of light, can grow to the size of cloud condensation nuclei and, accordingly, affect the meteorological processes in the atmosphere. This is especially important for a more

complete understanding of the physics of clouds.

**8 Comment #8**

**8.1 Reviewer comment**

*Has any thought been given to the addition of a photolytic source? Most NPF is driven by photolytic processes.*

**8.2 Response**

The article does not consider the chemical processes of the aerosol particle formations.

**9 Comment #9**

**9.1 Reviewer comment**

*I feel that if the experiments discussed were repeated with the inclusion of the above suggested measurements and the results discussed in terms of chemistry (oxidation, ozonolysis, formation of low volatility species, etc) this work would be a strong addition to AMT.*

[Figure]

**9.2 Response**

The literature broadly covers works that describe the chemical processes of the new particle formation.

In our work for the first time, we experimentally showed that new particle formations independently evolve in size. We provided experiments under conditions corresponding to the natural conditions of a real atmosphere. Particles enlarged in size can serve as a significant source of replenishment of the numerical concentration of cloud condensation nuclei.